# Photo-Assisted Removal of Rhodamine B and Nile Blue Dyes from Water Using CuO–SiO_2_ Composite

**DOI:** 10.3390/molecules27165343

**Published:** 2022-08-22

**Authors:** Muhammad Yaseen, Muhammad Humayun, Abbas Khan, Muhammad Idrees, Nasrullah Shah, Shaista Bibi

**Affiliations:** 1Department of Chemistry, Abdul Wali Khan University, Mardan 23200, Pakistan; 2Wuhan National Laboratory for Optoelectronics, School of Optical and Electronic Information, Huazhong University of Science and Technology, Wuhan 430074, China; 3Additive Manufacturing Institute, College of Mechatronics and Control Engineering, Shenzhen University, Shenzhen 518060, China

**Keywords:** binary nanocomposites, structural analysis, photocatalysis, catalytic degradation, kinetics investigation

## Abstract

Wastewater from the textile industries contaminates the natural water and affects the aquatic environment, soil fertility and biological ecosystem through discharge of different hazardous effluents. Therefore, it is essential to remove such dissolved toxic materials from water by applying more efficient techniques. We performed a comparative study on the removal of rhodamine B (RhB) and Nile blue (NB) from water through a catalytic/photocatalytic approach while using a CuO–SiO_2_ based nanocomposite. The CuO–SiO_2_ nanocomposite was synthesized through a sol–gel process using copper nitrate dihydrate and tetraethylorthosilicate as CuO and SiO_2_ precursors, respectively, with ammonia solution as the precipitating agent. The synthesized nanocomposites were characterized, for their structure, morphology, crystallinity, stability, surface area, pore size and pore volume, by using a scanning electron microscope (SEM), transmission electron microscope (TEM), energy dispersive X-ray spectroscopy (EDX), X-ray diffraction (XRD), Fourier transform infrared spectroscopy (FTIR) and Brunauer–Emmett–Teller (BET) techniques. The CuO–SiO_2_ nanocomposite was used for potential environmental applications in the terms of its catalytic and photocatalytic activities toward the degradation of rhodamine B (RhB) and Nile blue (NB) dyes, in the presence and absence of light, while monitoring the degradation process of dyes by UV-Visible spectroscopy. The catalytic efficiency of the same composite was studied and discussed in terms of changes in the chemical structures of dyes and other experimental conditions, such as the presence and absence of light. Moreover, the composite showed 85% and 90% efficiency towards the removal of rhodamine B and Nile blue dyes respectively. Thus, the CuO–SiO_2_ nanocomposite showed better efficiency toward removal of Nile blue as compared to rhodamine B dye while keeping other experimental variables constant. This can be attributed to the structure–property relationships and compatibility of a catalyst with the molecular structures of dyes.

## 1. Introduction

The vital compound for all living organisms is water, and that constitutes about 70 to 90% of their body weight [1]. Nowadays, due to industrial waste, water, especially drinking water and the aquatic environment, are being polluted by various chemicals. The dye contaminants having applications in leather, textile dyeing, pharmaceutical, paper printing, nutrition and cosmetic industries are the major sources of industrial derelict water. Wastewater from the textile industries is a mixture of organic, inorganic, polymeric and elemental pollutants. The water environment, aquatic creatures, soil fertility and biological ecosystems are contaminated by the different hazardous wastes of the various industries. Dyes are one issue from textile industries which are toxic to the biological environment by blocking the sun, resulting harmful impact on the ecosystem. Its contamination of the aquatic environment is greatest in comparison to other industries [2]. Dyes are of various types: basic, acidic, neutral, disperse, azo, direct, reactive, etc. A small quantity of dyes can affect the water environment, which is the reason the decolonization of wastes has a significant effect on the colour and organic substances. In order to keep the aquatic environment clean and conserved, the removal of these dyes from aqueous solutions is very important. Currently, various processes are in place for the treatment of waste water that holds various carcinogenic, non-biodegradable and hazardous derivatives [3,4]. The adsorption, degradation and decomposition phenomena are used for removal of colours from aqueous media. Treatment of the wastewater of food, paper, textile and dyeing industries before discharge to the aquatic environment is a challenge. Some of the various methods used for the removal of pollutants are: (1) Biological methods, whose function is to remove dye from wastewater with little cost. (2) Chemical methods, which are higher in cost and generate secondary hazardous byproducts. The various mechanisms involved in this process are electrochemical, advanced oxidation processes (AOPs), reduction, ozonation and the Fenton reaction [5]. (3) Physical methods, comprising flocculation, reverse osmosis, ion exchange, adsorption, irradiation, membrane filtration [6] biosorption, biodegradation, adsorption [7], ultrafiltration [8], nanofiltration [9], coagulation and sedimentation [10], the Fenton process [11], sonolysis [12], ozonation [13] and so on. Chemical removal of dyes is the most efficient method among all the mentioned processes, irrespective of its disadvantages. Recent reports showed that some of these conventional methods are not suitable for dye removal because of their high cost, time-consuming processing and stability problems. Although some of these methods are useful, they lead to secondary waste products requiring further reactions due to the stable and complex aromatic structures of dyes. There is a dire need for an efficient process which is relatively cost-effective and versatile and leads to no secondary pollutants, i.e., sludge.

Multistep processes have been advanced for the degradation of dyes. The degradation of dyes by metallic catalysts has gotten much consideration because of high proficiency and greener route of degradation. The various types of photocatalysts used for degradation of chemicals and pollutants include several semiconductors [14]. Some of the various semiconductors, such as ZnO [15,16], CeO_2_, MnO_2_, TiO_2_ [17], Cu_2_O [18], CdS [19], TiO_2_ [20] and Fe_2_O_3_ [21], have been used in photocatalytic applications. Remarkable features, such as high stability, selective electronic structure and low price, make CuO one of the more impressive photocatalysts [22]. Apart from the many great advantages of CuO nanoparticles, their use in the area of photocatalysis is still limited due to the accelerated electron-hole re-consolidation and the bandgap energy expansion; subsequently, a material with low efficiency would be produced. The photocatalytic ability of CuO particles could be enhanced through increasing the charge separation, and this could be achieved by its combination with other semiconductors to make composite materials.

Similarly, SiO_2_ was used in combination with CuO nanoparticles because the size and high reactivity of the surface area of SiO_2_ nanoparticles and their non-toxicity made them useful in various areas, such as catalysis, in biomedical and chemical sensors, chromatography and ceramics. The nanocomposites of SiO_2_ nanoparticles with other inorganic nanoparticles, such as Mn_2_O_3_-SiO_2_, showed good photocatalytic activities by using various dyes, such as crystal violet (CV) [23]. Some binary and ternary nanocomposite-based materials, such as the CoFe_2_O_4_ catalyst, are employed for the degradation of rhodamine B and methylene blue removal from the aqueous media [24]. Sharma and Basu have reported good catalytic performance of CuO@SiO_2_ monoliths for the degradation of rhodamine B (RhB) dye with more than 80% removal efficiency [25]. The Bi/SnO_2_/TiO_2_-G catalyst showed more than 60% efficiency toward the degradation of pentachlorophenol [26]. The C/SiO_2_ nanocomposite was employed for the removal of C.I. Basic Red 46 (BR46) dye [27]. B-SnO_2_ was used for the degradation of crystal violet and rhodamine B [28]. Likewise, we also prepared and used TiO_2_@MIL_53_Fe and CeO_2_@MIL_53_ Fe photocatalysts for the degradation of 2,4-dichlorophenol (2,4-DCP) [29] and CuO–SiO_2_ for the degradation of the crystal violet (CV) with degradation efficiency of 70% and 90% [23]. Fatimah et. al. studied the ZnO/SiO_2_ nanocomposite for the removal of RhB dye and found excellent decolourization efficiency of (99%) [30]. Some other catalysts, such as MIL53Fe@MIL53Sr, Mn/BiOCl, Mn/NiO, SiO_2_/CdO/CdS, Nd-SiO_2_-TiO_2_, TiO_2_@SiO_2_–Ag, TiO_2_-SnO_2_, ZnO/CdSe and SiO_2_-Fe_2_O_3_, were fabricated and employed for the photo-assisted degradation of various organic dyes and found that the composites showed appreciable efficiencies [31,32,33,34,35,36,37,38,39]. In addition to these, the titanosilicate/bismuth vanadate (BVTS) photocatalyst [40], Fe_3_O_4_@SiO_2_@TiO_2_/graphene oxide (GO) [41], ZnO-MnFe_2_O_4_ [42], γ-Fe_2_O_3_@SiO_2_@TiO_2_–Ag [43],γ-Fe_2_O_3_/SiO_2_ [44], γ-Fe_2_O_3_@SiO_2_@TiO_2_ [45], SnS_2_-SiO_2_@α-Fe_2_O_3_ [46], Fe_3_O_4_@SiO_2_@ZnO/CdS [47], Fe_3_O_4_@SiO_2_@ZnO@Au (FSZA) [48,49], FeNi_3_@SiO_2_@ZnO [50], Fe_3_O_4_@SiO_2_@TiO_2_–Sn [51], Fe_3_O_4_@SiO_2_@TiO_2_@Pt [52], etc.—all composite materials—were successfully fabricated, characterized and fruitfully employed for the photocatalytic degradation of various dyes, such rhodamine B, Basic blue 41, Nile blue, methylene blue (MB) and tamoxifen (TMX), in water-based media. A brief literature survey shows that the efficiency of the catalyst can be tuned by changing various experimental variables, the quality and quantity of pollutants, its chemical composition and various textural and morphological aspects; thus, the fabrication of a material with balanced behavior is the need of the day. Therefore, to achieve some of such objectives and in order to extend our work [23], we synthesized a CuO–SiO_2_-based nanocomposite through an economical and easily fabricated sol–gel process and successfully utilized it as an efficient catalyst for the removal of two structural different dyes, rhodamine B and Nile blue, from the aqueous media. Furthermore, systematic physiochemical and kinetic studies on the comparative degradation efficiency of the CuO–SiO_2_ composite towards the removal of two chemically different dyes were also carried out in detail. Moreover, the percentage removal efficiency of the catalyst under investigation was also compared with those of previously reported materials for the removal of same dyes.

## 2. Results and Discussion

### 2.1. UV-Visible, Bandgap Energy and FTIR

The UV-Visible analyses were performed by dispersing the nanocomposite in absolute ethanol while using UV-Visible Spectrophotometer Lamda-25 (PerkinElmer). These results of the synthesized composite are shown Figure 1a. The absorption peak observed at a wavelength of 300 nm confirmed the CuO–SiO_2_ composite’s formation. The other researchers have also gotten a peak at 256 nm for the corresponding composite [53]. Figure 1b indicates the indirect bandgap energy of the synthesized composite materials which was calculated using the following Taucs plot relationship.
(αhʋ)^γ^ = A(hʋ − Eg)(1)

Here, α = absorption coefficient, h = planks constant, ʋ = photon frequency, A = proportionality constant, Eg = bandgap energy and γ = electronic transition that may be 2, 1/2, 2/3 or 1/3, corresponding to the transition occurring. A straight line is obtained by plotting (αhʋ)^1/2^ versus hʋ for indirect transition. The intercept of the straight line on the hʋ axis shows the optical bandgap energy (Eg) of the composite. The indirect bandgap energy of the composite obtained was around 3.419 eV. The lower value of the bandgap energy for the composite is an indication of the bigger size of the particles of the composite, which may be attributed due to the incorporation of CuO in the polymeric silica matrix. The FTIR spectrum of the CuO–SiO_2_ composites was examined with a Perkin Elmer series 100 FTIR spectrometer with a resolution capacity of 5 cm^−1^, and the spectrum was recorded at 400–4000 cm^−1^. The peak detected at 450cm^−1^ was due to the CuO vibration**.** The bands corresponding to 803 and 1066 cm^−1^ were due to the Si-O-Si asymmetric stretching vibration, as indicated in Figure 1c. Thus, in addition to the UV-Visible spectroscopic results, it was also confirmed through the FTIR that the synthesized composite was CuO–SiO_2_.

### 2.2. SEM, TEM and EDX

To study the morphology of the composite, the SEM (JEOL JSM-7001F) results are shown in Figure 2a. These indicate the CuO particles were dispersed uniformly with the polymeric SiO_2_ particles. Similarly, for the study of internal morphology, transmission electron microscopy (TEM) was also performed, and the corresponding micrograph is shown in Figure 2b. The structural characterization of the synthesized composite, from SEM and TEM results, has also confirmed the formation of the synthesized composite. Further, it was also confirmed that the CuO particles were homogeneously dispersed in the SiO_2_ particles, leading to the big size of the composite materials. Similarly, Figure 2c indicates the energy dispersive X-ray spectroscopy (EDX) result of the synthesized composite materials. The results clearly demonstrate the elemental composition regarding Cu, O and Si of the materials, confirming the successful fabrication of CuO–SiO_2_ composite qualitatively and quantitatively.

### 2.3. XRD and TGA

The X-RD results of the synthesized composites of CuO–SiO_2_ and CuO particles are shown in Figure 3a. The diffraction peaks ascribed at various angles, 2θ, are 32.3° (020), 35.5° (200), 38.6° (20-2), 48.7° (022), 53.3° (120), 58.3° (221), 61.2° (22-2), 65.9° (134-2), 68.0° (0–24), 72.3° (123), 75.1° (125), 80.2° (220) and 82.7° (33-1), were observed in the spectra of both CuO particles and their SiO_2_-based binary composite. The only major difference in these both spectra was due to the presence of a broader peak at an angle of 23.7° corresponding to the SiO_2_ in the case of the CuO–SiO_2_ composite; minor shifts in the intensity and position of the original CuO peaks were also observed for the composite materials. All these facts not only confirm the successful formation of the CuO–SiO_2_ composite, but also reflect the semi-crystalline nature of composite materials. The crystalline behavior was due to the presence of CuO, and the presence of some amorphous phases was because of polymeric SiO_2_. The approximate crystallite size of the materials was determined using the sherrer equation.

(2)
$$D=\frac{K\lambda }{\beta \cos\theta }\;$$

where, D = average crystallite size (nm), K = Scherrer constant (0.89), λ = wavelength (1.54059 °A), β = FWHM (rad) and θ = Bragg’s angle (degrees). The percent crystallinity index was determined using the following formula and was found to be in the range of 9–10 nm.

The TGA result of the synthesized composite CuO–SiO_2_ is shown in Figure 3b. The synthesized composite showed weight loss at various thermal temperatures: 95, 647, 902 and 1010 °C. The first weight loss (0.32%) at 95 °C was because of the evaporation of physically adsorbed water and/or alcohol from CuO–SiO_2_ materials. The second weight loss (1.28%) at 647 °C was due to the dehydroxylation of single silanols (Si-O-H) of the synthesized composite, and then the final weight loss (1.67%) at 902 to 1010 °C can be attributed to the removal of oligomeric SiO_2_ chains and some phase changes in the materials. Beyond this temperature range, no weight loss was observed. Furthermore, the overall minor weight loss indicates appreciable stability of the composite materials.

### 2.4. BET Results

The synthesized nanocomposites’ pore structure, thickness and specific surface area were studied via nitrogen adsorption desorption isotherm. Figure 4a–d shows the nitrogen adsorption isotherm. It can be inferred from the figure that the composite had a type IV isotherm and a H_3_ type hysteresis loop for the relative pressure P/P_0_ in the range of 0.6 to 1 according to Brunauer–Deming–Deming–Teller (BDDT) classification. It can be seen in the Figure 4a,b that the nitrogen adsorption isotherm increases slowly in slope from P/P_0_ = 0.6; its linearity is enhanced and reaches P/P_0_ = 1.0, which is related to the adsorption/desorption process. The specific surface area estimated by Brunauer–Emmett–Teller (BET) for CuO–SiO_2_ was also calculated and found to be 134.5623 m²/g, as shown by the nitrogen adsorption study (Figure 4). The surface area, pore volume and pore size were 134.5623, 0.555197 and 16.60413 nm. These results reflect the finer size and appreciable pore size, and hence, it is a suitable material for the removal of pollutants from water [23].

### 2.5. Catalytic/Photocatalytic Study

The synthesized nanocomposite CuO–SiO_2_ was used for the catalytic and photocatalytic removal of rhodamine B and Nile blue dyes from aqueous media at various time intervals using the UV-Visible Spectroscopic technique. The photocatalytic degradation process was carried out using a wooden box in which the inner wall was coated with an aluminum sheet to prevent light dispersion. Additionally, the photocatalytic box was supplied with a 254 nm 220V-15W UV-lamp. The photodegradation of rhodamine B (RhB) and Nile blue (NB) by CuO–SiO_2_ nanocomposites was examined. The photodegradation studies were carried out in the presence and absence of light with the catalyst CuO–SiO_2_. The degradation studies were performed in such a way that before exposing it to light, the mixture was stirred for 30 min to bring about the adsorption–desorption equilibrium between the dye molecules on the catalyst surfaces. Aliquots of 10 mL was taken from the reaction solution after regular intervals of time, filtered and examined through UV-Visible spectrophotometer, and we noted the absorbance at λ_max_. The percentages of degradation of the rhodamine B and Nile blue were calculated through the given formula.

(3)
$$\mathrm{At}=\left(\frac{A_{i}-{\;A}_{t}}{m}\;\right)\times 100$$

where “A_i_” is initial absorbance and “A_t_” is final absorbance at a specific time interval.

To monitor the dye removal process kinetically, the following first-order equation was used for treating the data.
ln(C_t_/C_o_) = −kapp t(4)
or
ln(A_t_/A_o_) = −kapp t(5)
where C_t_ is the concentration of rhodamine B and NB dyes at time “t” and C_o_ is the concentration at time t = 0. Similarly, A_t_ is the absorbance at time t and A_o_ is the absorbance at time t = 0. k_app_ is the apparent rate constant of the reaction calculated from the slope of plots of ln(A_t_/A_o_) versus time of reaction for different reactions carried out.

The photodegradation summary of rhodamine B (RhB) and Nile blue (NB), in the absence of catalyst, is shown in Figure 5a–d. Figure 5a indicates the variation in absorbance at various time intervals. Figure 5 gives the development of degradation in terms of the ratio of absorbance (A_t_/A_o_) versus time, indicating that the light-assisted self-degradation of the RhB dye occurs by observing the quantitative and qualitative changes in spectra with the passage of time. Moreover, examined from the plot of percentage decomposition versus time in Figure 5b, percentage of degradation increased with the passage of time, and about 25% of the dye was removed in five hours. In the absence of catalyst, the photo-assisted self-degradation of rhodamine B and Nile blue under UV light irradiation was studied, and it was observed that about 25–35% of each dye was degraded by continuous irradiation of the dye solution for more than eight hours. In the presence of catalyst, almost 80–90% of dyes were degraded in a shorter time. Hence, it can be said that only light can also help in the removal of dye from water, but its removal proficiency is quite low relative to that in the presence of catalyst. Similarly, Figure 5c also indicates the plot of absorbance versus wavelength with various time intervals, which clearly indicates the degradation of the Nile blue dye with the passage of time in presence of light. Figure 5c shows the progress of degradation in terms of the ratio of absorbance (A_t_/A_o_) versus time, which shows that the degradation of the Nile blue dye occurred by observing the changes in spectra with the passage of time. Figure 5d is the plot of percentage of decomposition versus time, which signifies clearly that the percentage of decomposition increased with the passage of time, and also exhibits that 35% removal of dye occurred in five hours. Similarly, Appendix A indicate the typical plots of pseudo-first and pseudo-second order relations for the self-degradation data of both dyes. It was confirmed that the degradation data follow a pseudo-first-order equation to a good, acceptable degree, as exhibited by R^2^ = 0.98 and 0.99 for the first-order rate constants (K_1_ = 0.0010 and 0.0013 min^−1^) for RhB and NB, respectively.

#### 2.5.1. *Rhodamine B Removal*

The photocatalytic activity of the CuO–SiO_2_ nanocomposites, calcined at 600 °C for 3 h, was studied by investigating the degradation of the rhodamine B (RhB) using UV-Visible light and in the dark using 0.1g of catalyst at different time intervals. It was observed that the intensity of the absorption peaks of the RhB observed at maximum wavelength λ_max_ = 552 nm was decreased with the passage of time. Figure 6a and Figure 7a show the variation in the UV-Visible spectrum of RhB in the presence of 0.1g CuO–SiO_2_ composite under light and in the dark. These figures show that there occurred qualitative and quantitative changes in the absorbance over time examined through double beam UV-Visible spectrophotometer. It is observed from the graph that intensity/absorbance decreased with the increase in irradiation time. The spectra also show a blue shift. This happened due to successive degradation of RhB with time under irradiation. Similarly, Figure 6b and Figure 7b gives the progress of degradation in terms of the ratio of absorbance (A_t_/A_o_) over time, indicating that the degradation of the RhB dye with catalyst CuO–SiO_2_ occurred by observing the quantitative and qualitative changes in spectra with the passage of time, which was also examined in the plot of percentage decomposition versus time. Figure 6c and Figure 7c show that the percentage of degradation increased as time passed, and also shows that about 30% dye removal occurred in 5 h. Figure 6d and Figure 7a indicate the plots of the pseudo-first-order relation for the degradation of RhB in the presence and absence of light, respectively. It can be clearly seen that the degradation data follow the first-order rate equation closely, as shown by their R^2^ = 0.9756 and 0.9836 in light and dark conditions. The corresponding values of the first-order rate constant (K_1_), 0.0075 min^−1^ under light and 0.0059 min^−1^ in the dark, are given in Table 1. It can be seen in the results that removal of RhB dye is faster under light as compared to in the dark. Further, the CuO–SiO_2_ catalyst had good removal efficiency toward RhB under both experimental conditions used.

#### 2.5.2. Nile Blue Removal

Similarly, Figure 8a and Figure 9a are plots of absorbance changes versus wavelength at various time intervals for the removal of Nile blue (NB) dye in the presence of 0.1 g of CuO–SiO_2_ binary composite in the absence and presence of light, respectively. It is seen that the absorbance at (λ_max_ = 592 nm) regularly decreased with the passage of time, which confirmed the activities of the nanocomposite toward the removal of NB dye. Likewise, Figure 8b and Figure 9b indicate the degradation of dye in the terms of changes in the plot of ratio of absorbance (A_t_/A_o_) as a function of time. Figure 8c and Figure 9c are plots of the percentage of decomposition versus time, which exhibits clearly that the percentage of decomposition increased with time; however, the percentage of decomposition was greater under light (90%) as compared to that in the dark (75%). Similarly, Figure 8d and Figure 9d show the plot of pseudo-first-order kinetics for the degradation of Nile blue with catalyst CuO–SiO_2_. It was confirmed that the degradation data follow a pseudo-first-order equation well, as exhibited by R^2^ = 0.95833 in the absence of light and R^2^ = 0.98891 under light. It is seen that initially the degradation was faster, and after one hour the slope of the percentage of degradation vs. time profile became less steep; this shows that the availability of the active sites of the catalyst decreased with the passage of time. Some increase in the removal efficiency of the catalyst in the presence of light can also be attributed to activation of the catalyst by light and also to the increase in the vibrational energy of bonds between dye molecules; and ultimately, that resulted in relatively faster removal of dyes from aqueous media. To summarize the discussion, it was observed that the catalytic/photocatalytic removal of RhB and NB dyes by CuO–SiO_2_ was satisfactory, and these activities were further enhanced in the presence of light, as can be seen in the kinetic data/rate constant. Furthermore, Nile blue could be removed easily and quickly as compared to rhodamine B dye while using the CuO–SiO_2_ composite as the catalyst/adsorbent. This may be due to the difference in the chemical structures/natures of the dyes. RhB was relatively stable as compared to Nile blue in our experimental conditions.

Moreover, a comparison of the percentage removal efficiencies of the same catalyst toward these chemically different dyes showed that the removal efficiency of the catalyst is also affected by varying the chemical structure of the dye/substrate. These dyes have different chemical structures, and the results suggest that the removal efficiency of the CuO–SiO_2_ composite is higher for Nile blue than for rhodamine B; this depends on the compatibility of the structure of the dye with the morphological and texture properties, especially the response of the active sites to the attachment of substrate molecules to its surface. In this case, it can be seen that the lower removal quantity of RhB compared to NB, under the same experimental conditions, can be assigned to the bulkier molecular size and greater stability of RhB compared to NB. This behavior of RhB was also observed in the light-assisted self-degradation (without CuO–SiO_2_) studies as well. Table 1 also shows a comparison of percentage removal efficiencies of the present materials with those of some other previously reported composite materials for the degradation of the same dyes (RhB and NB). It is shown that the cheaper and more easily fabricated composite based on CuO–SiO_2_ has appreciable efficiency in comparison to other materials. A close observation of the results showed that the photocatalytic degradation efficiency in the various experiments varied with the experimental conditions, such as pollutant concentration, amount and type of catalyst, dose of catalyst, light source and chemical structure of the dye. More importantly, the efficiency of the catalyst is also affected by its chemical composition, and various texture and morphological aspects. As for the texture of the material, the differences in the particle size, shape, surface area, surface volume, porosity, arrangement and proportions of compositional parts of the material play vital roles. Therefore, it is important to develop and fabricate a material with well-balanced levels for all these characteristics; this will be helpful for the overall improvement of a single material for the catalytic purposes of different processes.

## 3. Experimental

### 3.1. Materials and Methods

Chemicals used in the experimental procedure were of analytical grades and were used without further treatment. These include copper nitrate trihydrate (Cu(NO_3_)_2_. 3H_2_O Sigma-Aldrich, St. Louis, MO, USA), tetraethyl orthosilicate (Si(OC_2_H_5_)_4_ Sigma-Aldrich, St. Louis, MO, USA), glycerol (C_3_H_8_O_3_ Sigma-Aldrich, St. Louis, MO, USA), nitric acid (HNO_3_ Sigma-Aldrich, St. Louis, MO, USA), ammonia solution (NH_3_ 30% Merck) and absolute ethanol (C_2_H_5_ OH Sigma-Aldrich, St. Louis, MO, USA). Rhodamine B (RhB) and Nile blue (NB) (purity, 99%) were acquired from Exciton (Dayton, OH, USA) and employed as obtained. The molecular formulae of rhodamine B and Nile blue are C_28_H_31_ClN_2_O_3_ and C_20_H_20_ClN_3_O, respectively, and their chemical structures are shown in Figure 1.

### 3.2. Preparation of CuO–SiO_2_ Composite

The CuO–SiO_2_ binary composite was prepared while following the previously published procedure, with slight modifications, via sol–gel method [23]. In this method, 40 mL of copper nitrate (1 M) solution was introduced in a 200 mL beaker, and to it 10 mL of glycerol was added dropwise under constant stirring at room temperature; and then 15 mL of TEOS was introduced dropwise, followed by the addition of 30% HNO_3_ in order to maintain the pH of the reaction mixture to 1.5 while stirring it for 1 h. In order to shift the pH of the reaction mixture from acidic to basic, the addition of 30% ammonia was performed dropwise until a pH ≈ 9 was obtained; and the reaction was allowed to occur for 3–4 h at 60 °C under continuous stirring. As a result, a blue gel was formed, which was then separated from the mixture through filtration; washed with water and then with ethanol more than three times; and kept in a water/ethanol (1:1) mixed solvent for 3 days. Finally, the mixture was filtered again and dried in an oven at 100 °C for 6 h, and then calcined at 600 °C for 3 h to get nanocomposite in pure powder form. A summary of the procedure is presented in Figure 2.

### 3.3. Characterization of the Synthesized Composite

The synthesized composite CuO–SiO_2_ was characterized by using various techniques: UV-Visible, SEM, TEM, EDX, FTIR, XRD, TGA and BET. The UV-Visible analysis was performed by dispersing the solid powder in the solvent and then subjecting the solution to UV-Visible Spectrophotometer Lamda-25 (PerkinElmer, Waltham, MA, USA) for analysis. SEM analysis was performed by using a scanning electron microscope, model JSM-5910 JEOL (Tokyo, Japan), to study the morphology of the synthesized nanocomposite. For the study of the internal morphology, the TEM analysis of the prepared composite was brought about using a JEM 2100F with a field-emission gun functioning at 200 kV. Similarly, the FTIR analysis was also carried out through FTIR (500–4000 CM1) with a Nicolet 1S5 from the USA with wave numbers 400–4000 cm^−1^, in which solid powder was injected for the analysis. The elemental composition of the synthesized CuO–SiO_2_ composite was determined through energy dispersive X-ray spectroscopy (EDX) using a JED-2300 analysis station with Acc. voltage 20 kV and probe current = 1.00000 nA. To check and analyze the crystallinity of the material, XRD analysis was performed (JDX-3532 JEOL, Tokyo, Japan, radiation source Cu K (alpha) 1.54 Angstrom). The diffraction data were recorded for the composite with an X-ray diffraction meter using Cu-K Alpha1 radiation with wave length (λ) equal to 0.154056 (1.54056 A°) in the range of 10–80° two theta (2Ө). To check the stability of the prepared composite, TGA analysis was also performed. Furthermore, for a specific surface area, pore size and pore volume, the BET analysis was also carried out.

### 3.4. Photocatalytic Degradation of the Dyes

The photocatalytic degradation method was carried out in a Pyrex glass beaker coated externally with aluminum foil to avoid light dispersion and focus it mainly on the reaction in the beaker. It was supplied with a UV lamp at about 200 nm, 220–15 W. In a typical process, 0.1g of the catalyst was dispersed in 100 mL of a 20 ppm-each rhodamine B and crystal violet solution in water in a Pyrex glass beaker with a magnetic stirring system. The degradation and photo-assisted degradation of rhodamine B and Nile blue dyes were carried in the presence of catalyst in the dark and under a light, respectively. Likewise, the stability and/or self-degradation of both dyes were also traced in the absence of catalyst under light, degraded by continuous irradiation of the dye solution for more than eight hours. At the start of each individual experiment, the dye solution was stirred for about 30 min in the dark to check the adsorption–desorption dynamic equilibrium between the dye molecules on the surface of the catalyst. A small amount of about 10 mL was taken at regular intervals and analyzed by noting the absorbance at λ_max_ through a double beam UV-Visible spectrophotometer. The values of absorbance, so obtained, were used to track the remaining concentration of dye in the solution. The data of absorbances/concentration were also tested by pseudo-first-order and pseudo-second-order kinetics models as well.

## 4. Conclusions

In this work, a binary nanocomposite of CuO–SiO_2_ was successfully synthesized using the sol–gel process and applied for the removal of two different dyes from aqueous media. The material so synthesized was characterized through spectroscopy, SEM, TEM, EDX, XRD, FTIR, TGA and BET techniques for its structure, morphology, crystallinity, thermal stability, pore size, surface area and pore volume. The composite was applied as a catalyst for the catalytic/adsorptive removal of Nile blue and rhodamine B dyes from aqueous media, and was found to be a suitable remover of the dyes for the aqueous system. The catalytic and photocatalytic activities against the two dyes were compared. The catalytic/photocatalytic activities of the composite CuO–SiO_2_ toward the removal of RhB and NB dyes were enhanced in the presence of light. Furthermore, the composite could remove NB easily and quickly as compared to RhB; and this is attributed to the difference in the chemical structures/natures of the dyes. The kinetic study indicated that the dye removal process follows a pseudo-first kinetic mechanism. In addition, a comparative study regarding the photo-assisted catalytic removal efficiency of the present catalyst with those of previously reported catalysts toward the removal of similar dyes showed that this inexpensive and easily fabricated catalyst has appreciable efficiency. The differences in the removal efficiencies of various materials are not only affected by their chemical compositions, but also by their various texture and morphological aspects. These include differences in the particle size, shape, surface area, surface volume, porosity and arrangement and proportions of their compositional parts. It is of utmost importance to have a material with well-balanced values of all these characteristics. It is also suggested that the creation of some charges/active sites on the surface of this composite may be useful for the efficient removal of ionic/inorganic pollutants from water through electrostatic interactions as well.

## Data Availability

Not applicable.

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
