# Peer review of "Photo-Assisted Removal of Rhodamine B and Nile Blue Dyes from Water Using CuO–SiO2 Composite"

_molecules, 2022, doi:10.3390/molecules27165343_

Round 1
Reviewer 1 Report
This manuscript reported the synthesis of CuO-SiO2 nanocomposite for photocatalytic degradation of Nile blue and Rhodamine B dye. The authors investigated the structure-property relationship and compatibility of the catalyst and dye molecular structures. The result is interesting. Overall, I would like to recommend its major revision, and the following issues should be addressed.
1. There are many literatures reporting that photocatalytic dyes degradation with excellent activity. Please explain the innovation of this work.
2. In Figure 2, HRTEM images should be provided to demonstrate the contact state of CuO and SiO2.
3. The standard XRD pattern of the reference material needs to be provided.
4. The stability performance for degradation of RhB and Nile blue should be provided.
5. There are a large number of inappropriate English words or expressions in the manuscript. The authors should carefully polish the English of the whole manuscript.
Author Response
Response/Reply to Reviewer’s Comments
First of all we would like to pay our special thanks to both editors and reviewers for helping us to make this work more acceptable/scientific. Please find our response/explanation to reviewer’s comments one by one. Thank you. Please note that all the changes made to the manuscript are also highlighted in Yellow Colour in the revised version.
Reviewer: 1
Comments to the Author
This manuscript reported the synthesis of CuO-SiO2 nanocomposite for photocatalytic degradation of Nile blue and Rhodamine B dye. The authors investigated the structure-property relationship and compatibility of the catalyst and dye molecular structures. The result is interesting. Overall, I would like to recommend its major revision, and the following issues should be addressed.
Response to Reviewer: 1
Thank you so much for recommending our work for publication in the “Molecules” after some fruitful changes/improvements. Please find our response/explanation to reviewer’s comments one by one.
Comment 1: There are many literatures reporting that photocatalytic dyes degradation with excellent activity. Please explain the innovation of this work.
Response 1: Thank you so much for your fruitful suggestions. Yes please you are right. Many researchers have studied various catalytic materials towards the photocatalytic dyes degradation with admirable activity. In most cases the materials and methods followed for the said purpose are bit expensive. However, in the present case we have applied well-known materials and synthesis with inexpensive and easily accessible approaches. The materials/methods followed in this work can be carried out in any less-advanced lab. This approach does not need highly skilled scientific personals. We do not say that this work is superior to that with excellent activities, yet it will give additional benefits to the field in terms of applied and academic point of view. This work also help in the addition of some part of new knowledge to the ultimate knowledge of this field, as we have performed a systematic physicochemical study of the comparative degradation of two structurally different dyes while using the same less expensive CuO-SiO2 binary composite. This specific comparative activity of this material can be generalized to other organic/inorganic pollutants as well. In addition we have also added some more data to Table-1 to show the % removal efficiency of the present materials in comparison to some other catalyst already reported for the same dyes RhB and NB in the revised version of manuscript. We hope this may be a fruitful addition to the field concerned.
Comment 2: In Figure 2, HRTEM images should be provided to demonstrate the contact state of CuO and SiO2.
Response 2: As per your kind suggestions a new TEM image with some better resolution is provided in the revised manuscript.
Comment 3: The standard XRD pattern of the reference material needs to be provided.
Response 3: Thank you. In the light of your valuable suggestions, the XRD pattern of CuO particles is also added in the XRD figure and the same is discussed in the text a well.
Comment 4: The stability performance for degradation of RhB and Nile blue should be provided.
Response 4: Thank you so much. In the light of yours advice a New Figure (Figure 5) is added in the revised manuscript to show the stability of of RhB and Nile blue dyes under the light and in the absence of catalyst. About 25-35% of the dye was degraded by continuous irradiation of the dye solution for more than Eight hours. While in the presence of catalyst almost 80-90% of dyes were degraded in a shorter time.
Comment 5: There are a large number of inappropriate English words or expressions in the manuscript. The authors should carefully polish the English of the whole manuscript.
Response 5: Thank you so much for your fruitful suggestions. In the light of your valuable suggestions the WHOLE manuscript was revised carefully and tried to avoid any English/Grammar syntax errors. We believe that the language may be now acceptable for the review process.
We are once again highly grateful to the reviewers and editors for helping us to make this work more attractive/useful for the readers.
Reviewer 2 Report
This paper is an interesting work about CuO-SiO2 composite for dye degradation. The materials were prepared by solvothermal approach and investigated for Rhodamine B and Nile Blue degradation, as informed in the title and abstract. Catalysts were characterized by UV-Vis, XRD, TGA, SEM and BET, and the photocatalysis studied by UV-Vis spectroscopy. The subject addressed in the manuscript is interesting and important, however the manuscript suffers basic mistakes and is not based on sufficient discussion and literature revision. In detail, my comments are listed as follows.
General:
Standardize text, nomenclatures and acronyms. For example, the word Sol-gel appears in different forms. Page 4, we can find XRD and X-RD. The authors use Figure and fig.
Improve figure captions.
Improve description of methodology and results. For example, the first paragraph in 2.2 item must be rewritten.
The manuscript must be carefully revised.
Results and discussion:
Figure 1 are wrong numbered (inserted graphs).
Item 2.2: Could the authors provide a EDX or XRF analysis to prove the presence of copper and silicious?
Item 2.5: The authors comment about crystal violet degradation (page 6, line 206), however the author showed results for nile blue.
Experimental:
Page 13, line 348: the authors cite crystal violet instead of nile blue. I really do not know which dye was studied.
Conclusion:
The authors should improve the conclusion section. For example, they could compare photocatalysis results with morphological and textural properties.
Author Response
Response/Reply to Reviewer’s Comments
First of all we would like to pay our special thanks to both editors and reviewers for helping us to make this work more acceptable/scientific. Please find our response/explanation to reviewer’s comments one by one. Thank you. Please note that all the changes made to the manuscript are also highlighted in Yellow Colour in the revised version.
Reviewer: 2
Comments to the Author
This paper is an interesting work about CuO-SiO2 composite for dye degradation. The materials were prepared by solvothermal approach and investigated for Rhodamine B and Nile Blue degradation, as informed in the title and abstract. Catalysts were characterized by UV-Vis, XRD, TGA, SEM and BET, and the photocatalysis studied by UV-Vis spectroscopy. The subject addressed in the manuscript is interesting and important, however the manuscript suffers basic mistakes and is not based on sufficient discussion and literature revision. In detail, my comments are listed as follows.
Response to Reviewer: 2
Thank you so much for your valuable suggestions and comments for the improvement of our work/manuscript. Thank you so much for considering our work for possible publication in the “Molecules” after some fruitful changes/improvements. Please find our response/explanation to reviewer’s comments one by one.
General:
Comments 1: Standardize text, nomenclatures and acronyms. For example, the word Sol-gel appears in different forms. Page 4, we can find XRD and X-RD. The authors use Figure and fig.
Improve figure captions.
Improve description of methodology and results. For example, the first paragraph in 2.2 item must be rewritten.
The manuscript must be carefully revised.
Response 1: Thank you so much for your so nice suggestions. As per your valuable suggestions the WHOLE manuscript was revised carefully and tried to avoid any English/Grammar syntax, nomenclatures and acronyms errors. Figure captions, overall methodology and discussion are now improved as well. It is hoped that the revised manuscript is now improved and be acceptable for the review process.
Results and discussion:
Comments 2: Figure 1 are wrong numbered (inserted graphs).
Item 2.2: Could the authors provide a EDX or XRF analysis to prove the presence of copper and silicious?
Item 2.5: The authors comment about crystal violet degradation (page 6, line 206), however the author showed results for nile blue.
Response 2: Thank you so much for your fruitful suggestions. Necessary changes to figure 1 are added. The EDX spectrum of CuO-SiO2 nanocomposite is now incorporated in the revised manuscript which can be provide some more information regarding the qualitative and quantitative information of the composite.
In addition the corrections regarding CV and NB are fixed as well. Thank you once again.
Experimental:
Comments 3: Page 13, line 348: the authors cite crystal violet instead of nile blue. I really do not know which dye was studied.
Response 3: We are really sorry and are much grateful to you for pointing out this mistake. The necessary corrections are incorporated in the revised manuscript.
Conclusion:
Comments 4: The authors should improve the conclusion section. For example, they could compare photocatalysis results with morphological and textural properties.
Response 4: Thank you respected professor for your so valuable and scientific suggestions for the improvement of our work. The whole manuscript including methodology, discussion, and conclusions parts are revised to a greater extent. We hope that our efforts in the light of yours comments might improve the revised version.
We are once again highly grateful to the reviewers and editors for helping us to make this work more attractive/useful for the readers.
Reviewer 3 Report
The manuscript deals with photocatalytic properties of CuO-SiO2. The authors fabricated CuO-SiO2 by sol-gel process and measured the photocatalytic dye degradation efficiency of the prepared CuO-SiO2. The authors also provided a lot of analyzed data to prove prepared materials were CuO-SiO2. However, the MS has lacks evidence to prove the authors’ claims. Thus, I could not recommend the MS to be published to Molecules.
For examples;
1) Although the absorbance in Figure 1 (a) showed the similar data to the data in the reference, the absorbance alone can not prove the prepare materials is CuO-SiO2. The authors should provide more clear evidence. In this sense, the authors should provide HR-TEM instead of Figure 2 (b).
2) The author should provide annotations of all the peaks of FTIR, XRD, and TGA.
3) The condition of materials preparation in experimental and results section is different, e.g., calcination condition of CuO-SiO2 is 500 ℃ for 4 h in experimental part and 600 ℃ for 3 h in results section.
4) To compare dye degradation data, the author should provide self-degradation data of dyes. (Without CuO-SiO2, under light)
5) The author would rewrite introduction to be more correlated to the main contents of the MS.
Author Response
Response/Reply to Reviewer’s Comments
First of all we would like to pay our special thanks to both editors and reviewers for helping us to make this work more acceptable/scientific. Please find our response/explanation to reviewer’s comments one by one. Thank you. Please note that all the changes made to the manuscript are also highlighted in Yellow Colour in the revised version.
Reviewer: 3
Comments to the Author
The manuscript deals with photocatalytic properties of CuO-SiO2. The authors fabricated CuO-SiO2 by sol-gel process and measured the photocatalytic dye degradation efficiency of the prepared CuO-SiO2. The authors also provided a lot of analyzed data to prove prepared materials were CuO-SiO2. However, the MS has lacks evidence to prove the authors’ claims. Thus, I could not recommend the MS to be published to Molecules.
For examples;
Response to Reviewer:3
Dear professor we are very much grateful to you for giving us valuable suggestions and comments for the improvement of our work/manuscript. In the light of suggestion/comments from yours side as well as other learned reviewers, we have carefully revised and improve the manuscript; hence, we are hopeful for positive response from your side as well. All the changes/improvements incorporated in the revised version are highlighted. Also find our response/explanation to yours comments one by one please.
Comment 1: Although the absorbance in Figure 1 (a) showed the similar data to the data in the reference, the absorbance alone cannot prove the prepare materials is CuO-SiO2. The authors should provide more clear evidence. In this sense, the authors should provide HR-TEM instead of Figure 2 (b).
Response 1: Thank you so much for your valuable advice. As per your kind suggestions a new TEM image with some better resolution is provided in the revised manuscript. In addition, EDX spectrum is also added to the revised manuscript for more information regarding the elemental compositions of the material.
Comment 2: The author should provide annotations of all the peaks of FTIR, XRD, and TGA.
Response 2: Thank you. The necessary and suggested annotations of all the peaks are incorporated in the revised manuscript.
Comment 3: The condition of materials preparation in experimental and results section is different, e.g., calcination condition of CuO-SiO2 is 500 ℃ for 4 h in experimental part and 600 ℃ for 3 h in results section.
Response 3: Thank you so much for pointing out this typographic mistake. The necessary corrections are incorporated in the revised manuscript.
Comment 4: To compare dye degradation data, the author should provide self-degradation data of dyes. (Without CuO-SiO2, under light)
Response 4: Thank you so much for your fruitful suggestions. As per your worthy suggestion a new Figure (Figure 5) is added in the revised manuscript. This is showing the self-degradation data of dyes (RhB and NB without CuO-SiO2 and under the light). It was observed that about 25-35% of the dye was degraded by continuous irradiation of the dye solution for more than Eight hours. While in the presence of catalyst almost 80-90% of dyes were degraded in a shorter time.
Comment 5: The author would rewrite introduction to be more correlated to the main contents of the MS.
Response 5: Thank you so much. In the light of your fruitful suggestions the introduction part has re-written to make is more correlated to the main topic of the manuscript. In addition, the WHOLE manuscript was revised carefully and tried to avoid any English/Grammar syntax errors. It is hoped that the revised version may be better compared to the previous version.
We are once again highly grateful to the reviewers and editors for helping us to make this work more attractive/useful for the readers.
Round 2
Reviewer 1 Report
Given that the author has addressed the raised issues, I think it can be published in the current version
Reviewer 2 Report
Dear authors,
Thank you to improve the manuscript and follow all suggestions. I consider the paper adequate.
Best regards.